Efficient video face recognition based on frame selection and quality assessment

Kharchevnikova Angelina 1 angelina.kharchevnikova@gmail.com
http://orcid.org/0000-0001-6196-0564 Savchenko Andrey V. 2 avsavchenko@hse.ru
1 HSE University , Nizhny Novgorod , Russia
2 HSE University, Laboratory of Algorithms and Technologies for Network Analysis , Nizhny Novgorod , Russia
Ben Xianye
Electronic publication date: 2021 Feb 25
Publication date: 2021
Volume: 7
Electronic Location ID: e391
Received 2020 Oct 30; Accepted 2021 Jan 22
Copyright: © 2021 Kharchevnikova and Savchenko
Copyright year: 2021
Copyright holder: Kharchevnikova and Savchenko
License: This is an open access article distributed under the terms of the Creative Commons Attribution License, which permits unrestricted use, distribution, reproduction and adaptation in any medium and for any purpose provided that it is properly attributed. For attribution, the original author(s), title, publication source (PeerJ Computer Science) and either DOI or URL of the article must be cited.
License URL: https://creativecommons.org/licenses/by/4.0/

Keywords: Face recognition, Key frame selection, Knowledge distillation, Face quality assessment

Funding: Russian Science Foundation (RSF) 20-71-10010 The research is supported by RSF (Russian Science Foundation) grant 20-71-10010. The funders had no role in study design, data collection and analysis, decision to publish, or preparation of the manuscript.

==============================
The article is considering the problem of increasing the performance and accuracy of video face identification. We examine the selection of the several best video frames using various techniques for assessing the quality of images. In contrast to traditional methods with estimation of image brightness/contrast, we propose to utilize the deep learning techniques that estimate the frame quality by using the lightweight convolutional neural network. In order to increase the effectiveness of the frame quality assessment step, we propose to distill knowledge of the cumbersome existing FaceQNet model for which there is no publicly available training dataset. The selected K-best frames are used to describe an input set of frames with a single average descriptor suitable for the nearest neighbor classifier. The proposed algorithm is compared with the traditional face feature extraction for each frame, as well as with the known clustering methods for a set of video frames.

Introduction

Today advanced technologies in the field of biometric identification are becoming increasingly popular in various areas of public life. Existing identification systems use such features as voice, human pose, gait (Ben et al., 2019), etc. However, the best recognition quality is known to be obtained through face recognition techniques. Indeed, a face is one of the most reliable identifier of a person that is impossible to lose or forget. Hence, the face recognition has become widely used in secure companies, for example, banks, in order to prevent violations or to provide targeted advertising to customers. Major airlines are beginning to introduce similar technologies to identify passengers, taking the face image as an entrance ticket to the flight. Several telecommunication companies are equipping the latest gadget models with the face geometry authentication technology to protect personal data (Truong, Graf & Yanushkevich, 2019).

The general goal of a face identification task is to associate an input sequence of video frames {X(n)},n=1,N¯ with one of L subjects (classes) from the reference gallery (Zhao et al., 2003). The frame number here is denoted by n, and the total number of video frames is N. Suppose, classes are defined by using reference facial still images (photos) {Xi}, i = 1,2, …, L with a known label. To simplify the task, we assume that the considered video sequence contains frames of only one person with a previously detected face area (Kharchevnikova & Savchenko, 2018). Thus, the problem of face identification by video is an example of the multi-class classification task. However, an observed object here is not a single image, but a set of images (frames), so that aggregation methods should be used to compute a single descriptor of the whole video or combine the decisions for each frame (Kharchevnikova & Savchenko, 2016, 2018).

As a matter of fact, real-time face recognition systems are based on the analysis of frames received at a given frame rate. Therefore, the reliability and accuracy of these technologies directly depend on the quality of the images coming to the input of the algorithm. However, the environment is not always conducive to obtaining frames of good quality due to lighting conditions, low resolution of the video camera, face positioning, the presence of blur, etc., which leads to unstable work of the system and errors occurring (Chen & Zhao, 2019). Often, just a few key images from the incoming video sequence are enough for the algorithm to guarantee that a person on a video belongs to one of the subjects in the reference database (Savchenko, 2016).

Moreover, face recognition based on traditional deep convolutional neural networks (CNN) is rather slow due to expensive inference operation (Liu et al., 2019). Hence, it usually requires high-performance servers with graphics processors (GPUs) for real-time video recognition. Traditional approach with extraction of facial features (embeddings) from each input frame using very deep CNN significantly slows down the speed of decision-making, especially if the system performs real-time calculations on platforms limited by power and memory resources. Moreover, it is impossible to apply very complex CNN (Hernandez-Ortega et al., 2019) to estimate quality of each frame due to the same restrictions of real-time processing. Therefore, the problem of increasing the robustness and performance of face identification algorithms by video sequence remains an urgent topic in the field of computer vision and machine learning.

The main contribution of this paper is applying knowledge distillation from a cumbersome FaceQnet (Hernandez-Ortega et al., 2019) to a fast CNN in the face identification problem based on selecting K-best frames from an input video sequence. It is experimentally demonstrated that the proposed approach either improves the total running time or demonstrates the high accuracy (up to 10%) compared to the baseline aggregation of all frames with the ResNet-50 (VGGFace2) and InsightFace models. Several lightweight CNN models were trained in order to more effectively implement the face quality assessment. The knowledge distillation is used to reach the performance of slow quality assessment FaceQnet in fully unsupervised manner by using any large facial dataset without need for the training dataset with a given facial quality that is not distributed by Hernandez-Ortega et al. (2019).

Related Work

One of the key steps in solving the face recognition problem is to obtain image feature vectors to perform further classification. It is important to highlight that the deep CNNs can be used as a robust facial descriptor if they have been previously trained on an external very large facial dataset of celebrities (Cao et al., 2018). The study of Taigman et al. (2014) proposed the DeepFace architecture for face verification, that was trained on more than four million face images. The described model achieves an accuracy of 97.35% on the Labeled Faces in the Wild (LFW) dataset (Learned-Miller et al., 2016). As an extension of DeepFace (Taigman et al., 2015) used a semantic bootstrapping method to identify an effective training subset of images. In order to increase recognition accuracy, an ensemble of twenty-five CNN models is created by Sun et al. (2014), each of the model is aimed at identifying various local patches of a facial image. This solution achieves a 99.47% performance on the LFW dataset.

Several studies in the field of face identification are aimed to optimizing the loss function in the training of CNN models. Hence, Schroff, Kalenichenko & Philbin (2015) proposed the triplet loss function to train the FaceNet descriptor. Its training is carried out on a huge data array consisting of two hundred million images of faces belonging to eight million different people. The result is an accuracy of 99.63% on the LFW. One of the best solutions today is the training approach using the Additive Angular Margin Loss (ArcFace) loss function (Deng et al., 2019) that can achieve the accuracy of 99.82% on the LFW dataset.

In this article we consider several contemporary CNN architectures for face feature extraction, namely:ResNet-50 trained on VGGFace2 dataset (Cao et al., 2018),

MobileNet (Savchenko, 2019a) architecture has a significant advantage in size as well as recognition speed, which allows recognition on mobile devices (hereinafter “MobileNet-VGG2”),

InsightFace (ArcFace) is a deep architecture based on ResNet. The network was trained using the ArcFace loss function (Deng et al., 2019) using the MS-Celeb-1M dataset in which the images were previously normalized and converted to the MXNet binary format (Guo et al., 2016).

The process of inference in deep CNNs to obtain facial feature vectors from a frame of images is an expensive operation and requires powerful GPUs. Video analysis of a huge number of frames can take tens of seconds that significantly affects the overall performance of the recognition software. Also, due to the influence of external factors, such as illumination, loss of focus by a video device, strong displacements of the human head, etc., decision-making for each frame leads to errors and unstable work of the identification algorithm (Nasrollahi & Moeslund, 2008). Therefore, in this study we reduce the problem of facial recognition by video sequence to the selection of the key frames.

Recent works on key frames selection can be divided into three categories: algorithms based on clustering, optical systems, and quality assessment methods (Dhamecha et al., 2016). Techniques of the first group analyze a set of facial images relative to their distribution formed by feature vectors. The key features are the centers of the clusters, found using some clustering algorithm, for example, K-means (Hadid & Pietikainen, 2004). Methods of the second category select the main frames in accordance with the displacement of objects from frame to frame, which is extracted by the optical flow method, such as the Lucas-Canada approach (Saeed & Dugelay, 2010). Despite the popularity of these techniques, the algorithms are quite time-consuming.

Hence, today, researchers are mostly interested in methods of the third kind, that predict the quality of each frame. The search for high-quality video images requires small computational costs, therefore, the algorithms of this group can be integrated into even embedded real-time facial identification engines. The fundamental question in this approach is understanding of how to define the frame quality. For example, distortions from image compression and transmission algorithms involve the presence of undistorted reference images, for which a decision is made. However, the task becomes more complex when a newly entering frame is estimated “blindly” based on certain characteristics. Some papers take the human perception of quality as a basis (Ferrara et al., 2012; Best-Rowden & Jain, 2018). Others rely on calculating the correlation between the input image and the expected system recognition accuracy (Alonso-Fernandez, Fierrez & Ortega-Garcia, 2011).

The image quality assessment approaches differ in terms of characteristics extracted from the frame. In fact, the quality of the facial image can be affected by the face itself, its posture, expression, or the characteristics of the video camera, such as focusing, resolution, sharpness, etc. All these factors can be considered using traditional methods of digital image processing or algorithms based on deep learning technologies. For example, four quality metrics: face symmetry, sharpness, contrast, and image brightness are proposed by Nasrollahi & Moeslund (2008). One of the most significant factors affecting image quality is its brightness. This parameter is calculated based on the intensity value in each pixel using the Luma (Y) standard: (1) Yt=0.2126⋅R+0.7152⋅G+0.0722⋅B

(2) bt=Yt255⋅h⋅w

where h and w are height and width of the incoming frame X(n).

It is also possible to determine the quality of frames by calculating the image contrast (Nasrollahi & Moeslund, 2008): (3) ct=Yt2h⋅w−bt2

However, these methods require to define empirically a threshold considering which the frame should be related to the key one. This problem was solved by Chen et al. (2014), who offered a special technique for ranking the metric weights to adapt to any face recognition system. Even though this approach made it possible to more accurately determine key frames, it is still inferior to algorithms based on deep learning (Nasrollahi & Moeslund, 2008; Qi, Liu & Schuckers, 2018a, 2018b).

Moreover, there exist CNN-based methods for image quality assessment. For instance, the FaceQNet model for obtaining numerical characteristics of image quality is described by Hernandez-Ortega et al. (2019). This architecture is based on the deep ResNet-50, from which the last classification layer was extracted (He et al., 2016). Additionally, two new layers have been added to the FaceQNet model for regression. This network is trained on a subset of three hundred individuals of the VGGFace2 dataset. The main task of our work is to evaluate the quality of frames using quick and simple algorithms in order to subsequently accelerate the time of the face recognition system. Unfortunately, the FaceQNet model cannot significantly increase the efficiency of the face recognition process since this architecture is based on the heavy ResNet-50.

Proposed Approach

Frame quality assessment

In this article we use a lightweight CNN for face quality assessment. At first, let us follow the FaceQNet pipeline (Hernandez-Ortega et al., 2019), taking as a basis a similar dataset. Unfortunately, its authors did not provide the dataset on which the FaceQNet model was trained. Therefore, we propose here to consider the learning of an efficient network based on the knowledge distillation from FaceQNet. As student models, it is proposed to consider MobileNet (Howard et al., 2017), as well as a simple network of the LeNet type, consisting of only 4–5 convolutional layers (hereinafter FaceQNet mobile and FaceQNet light, respectively). Since the output of original FaceQNet (Hernandez-Ortega et al., 2019) is regression of the image quality value, the last layer of light networks is also the layer responsible for regression.

Our learning process with the knowledge distillation approach is shown schematically in Fig. 1. A subset of VGGFace2 (Cao et al., 2018), consisting of three hundred first classes, is considered to be a training set. This set is unlabeled in terms of facial quality, so that we train our network in completely unsupervised manner. Hence, at the first stage, it is necessary to obtain predictions of the quality from the teacher model (pre-trained FaceQNet) (Hernandez-Ortega et al., 2019) for each facial image from the training set. These estimates together with corresponding image make up a training set for the student model. As a student model, two architectures are under consideration: the modification of MobileNet (Savchenko, 2019a) and lightweight LeNet-based model. During the training, each image of the training sample is fed to the input of a light network, which returns predicted quality. The value obtained by a student model is used in the loss function, namely, the mean square error (MSE) is calculated based on reference estimates of facial quality provided by a teacher model. Finally, the weights of the student model are updated using the variation of stochastic gradient descent. The Adam optimizer was applied with learning rate 0.001 and decay 1e−5. The student models were trained in 10 epochs with early stopping based on the MSE on validation subset (20% of the whole dataset). Figure 2 demonstrates examples of images with the quality assessments obtained by the FaceQNet teacher model (Q0), as well as by the student models FaceQNet mobile (Q1) and FaceQNet light (Q2).

Figure 1 Distillation training pipeline.

Figure 2 Quality scores FaceQNet, FaceQNet mobile, FaceQNet light.

(A) High quality face image, (B) Low quality face image.

In addition, we train the lightweight CNN using the FIIQA dataset with associated lighting quality metrics (Zhang, Zhang & Li, 2017). Considering the proposed FIIQA-based approach the MobileNet architecture (Savchenko, 2019a), previously trained on VGGFace2 (Cao et al., 2018), is used as the basis for fine-tuning. A fully connected layer with Softmax activation function is added for classification, so that the categorical cross-entropy is considered as a loss function. The model was trained in 20 epochs with early stopping using the Adam optimizer. The output of the trained MobileNet FIIQA model is the likelihood that the image belongs to one of Q = 3 quality classes P*(q|X(n)), q ∈ {1,…,Q}. The final decision on the quality of the frame is interpreted using an estimate of the mathematical expectation (Liu et al., 2019): (4) bn=1Q∑q=1QP(q|X(n))⋅q

Proposed pipeline

Figure 3 demonstrates the pipeline of the face identification system with additional step of lightweight CNN-based frame quality assessment. Here the proposed block is highlighted in bold. At first, individual frames are extracted in the incoming video sequence with a fixed frame rate and pre-processed (e.g., normalized). Next, the face region is detected using the MTCNN (Zhang et al., 2016). The resulted facial regions are fed to the input of the stage of calculating the quality of the frames, where frame-by-frame evaluation occurs using one of the methods for analyzing the image structure (1)–(3) or using deep learning technologies from previous subsection. Next, we select the best frames for supplying to a CNN-based facial feature extractor. Let Q(X(n)) is the frame quality estimate from the input video sequence. We sort the obtained frame quality estimates in descending or ascending order (depending on the algorithm), from “best” to “worst”, so Q(X(1)) ≥ Q(X(n)) ≥ Q(X(N)). From the indices (1)…(N) we take the top-k, which we consider to be the key frames, where the hyper-parameter K ≤ N is determined empirically. The resulting K-best video images are the input to the CNN. The result of the block is the facial feature vectors of the frames. In order to get a single vector that describes the input video, we compute the arithmetic mean of features of each frame (Kharchevnikova & Savchenko, 2016). The final solution is made using the 1-NN descriptor (Nearest Neighbor) with Euclidean distance. This pipeline has been implemented in a publicly available Python application by using Keras framework with TensorFlow backend. Sample screenshot is shown in Fig. 4.

Figure 3 Proposed pipeline of video face identification.

Figure 4 Sample screenshot of the developed implementation of the proposed pipeline.

Experimental Results

The experiments have been conducted using two datasets:IJB-C (IARPA Janus Benchmark-C) (Maze et al., 2018) is one of the most popular face recognition datasets. The database contains 3,531 unique objects, namely the faces of celebrities, athletes, political figures for whom individual images and short videos have been collected. In total, the set contains 21,956 photographs of recognized classes, as well as 19,593 videos with pre-selected frames in the amount of 457,512. The average number of frames per video is approximately equal to 33 images. IJB-C contains many images with faces that are truly difficult to recognize.

YTF (YouTube Faces) dataset (Wolf, Hassner & Maoz, 2011) consists of 3,425 videos collected from the famous YouTube platform. Each video clip contains 181.3 frames in average. Recognition classes from YTF have an intersection of 596 subjects with static images from the LFW dataset (Learned-Miller et al., 2016).

Since the standard protocol for the IJB-C dataset (Savchenko, Belova & Savchenko, 2018) contains samples both from video frames and single images, it is not applicable for the current article. Thus, all the results in this article have been obtained from conducted experiments based on the protocol, where the training set contains only still images and the testing set consists of the videos only.

Running time

The efficiency of video-based face recognition largely depends on the speed of CNN together with methods of selecting high-quality frames. The analysis of algorithms performance is conducted on the AMD Ryzen Threadripper 1920X 12-Core Processor server, a 64-bit Ubuntu 16 operating system, RAM 64 GB, with Nvidia GeForce GTX 1080 Ti GPU. Table 1 presents the sizes of the pre-trained CNNs, as well as the average inference time tinference for one frame by CNN. In all tables, the best results are shown in bold and the worst results are marked by italics.

Table 1 Performance of CNN models for facial feature extraction.

CNN	Average inference time per frame tinference, ms	Model size, MB	
	GPU	CPU		
ResNet-50 (VGGFace2)	9.186	48.917	93	
MobileNet-VGG2	6.574	20.246	12.7	
InsightFace	15.735	90.407	170	
Note:

The best results are shown in bold and the worst results are marked by italics.

The MobileNet model here is the fastest of the proposed options in connection with the specifics of this network. The architecture optimized for mobile platforms is capable of processing 3–5 frames per second. MobileNet also has the advantage of the size of a trained balance, the difference is almost 14 times compared to the heavy InsightFace. On average, the GPU gives an acceleration of 5 times.

However, traditional face recognition algorithms involve inference in a CNN for each frame in the incoming video sequence. Following simple arithmetic calculations, one can notice an obvious linear increase in the overall running time T = N · tinference of the face identification system, depending on the growth in the number of incoming frames N. So, a video of N = 1,000 frames in length is processed in 8 s by the fastest of the described CNNs, namely, the multi-output MobileNet (Savchenko, 2019a).

It is worth noting that the key idea of our approach (Fig. 3) is applying effective algorithms for choosing K high-quality frames from the input video sequence. Then, the calculation of the total running time is reduced to: (5) T∗=N⋅tframe+K⋅tinference

where tframe is the time to estimate quality of a single frame. To increase the performance of the face recognition system, the running time of the algorithms for selecting high-quality frames should not exceed the time of CNN inference: tframe < tinference. The results of measurements of the frame quality assessment running time tframe per one image are presented in Table 2. The overall running time (5) for the quality assessment methods in dependance on the number K of selected frames are presented in Figs. 5 and 6. The fixed number of quality images K are fed to the input of the CNN (K = N/4 of the total number of all frames in the one video). Here, the traditional approach of facial extraction from each frame without facial quality assessment is marked as Baseline.

Table 2 Quality assessment methods performance.

Quality assessment	Average processing time per frame tframe, ms	Model size, MB	
	CPU	GPU		
Luminance (1), (2)	0.027	–	0	
Contrast (3)	0.071	–	0	
FaceQNet	12.039	47.897	93.33	
FaceQNet mobile	6.351	22.088	12.94	
FaceQNet light	3.111	14.213	4.4	
FIIQA mobile	6.642	21.409	12.83	

Figure 5 The inference time (CPU) depending on the number of frames, ResNet-50 (VGGFace2).

Figure 6 The inference time (GPU) depending on the number of frames, ResNet-50 (VGGFace2).

Here one can notice that a significant difference in the performance is observed if the number of frames is greater than 150. As soon as the number of frames increasing, simple and fast algorithms for selecting key video frames show their advantage. Thus, the most effective and less resource-intensive algorithms are simple methods for assessing Brightness (Luminance) and Contrast (Contrast). Approaches for selecting quality frames using lightweight CNN models have also shown their effectiveness (Figs. 5 and 6). The original FaceQNet model (Hernandez-Ortega et al., 2019) offers no advantage in recognition speed over pre-trained CNNs for feature extraction.

Accuracy

The previous subsection demonstrated that the proposed approach with distilled CNN (FaceQnet mobile) is able to improve the running time of video face recognition. However, our main goal is to increase the face identification accuracy by selecting only top-k best frames from the video sequence. Accuracy is defined as the ratio of correctly defined classes to the total number of predictions received by the 1-NN descriptor. All considered algorithms for selecting high-quality frames are compared with the traditional approach of identifying each frame (hereinafter referred as Baseline). The main objective of the study is to obtain recognition accuracy not lower than Baseline. Therefore, preference is given to methods with a sufficiently high proportion of correctly predicted classes. In order to compare the various approaches to search for key video images, several clustering algorithms are considered: K-means and MiniBatchKMeans. For the integrity, a predetermined number of images are randomly selected from the incoming sequence of frames and fed to the input of the face identification stage.

Tables 3–5 demonstrate the dependance of the accuracy on the number K of selected key frames for the IJB-C dataset using ResNet-50 (VGGFace2) (Cao et al., 2018), multi-output MobileNet (Savchenko, 2019a) and InsightFace (ArcFace) (Deng et al., 2019) facial descriptors, respectively.

Table 3 Recognition accuracy for IJB-C dataset, ResNet-50 (VGGFace2).

Algorithm	Number of selected frames K	
N/2	N/4	N/8	2	1	
Baseline		70.892	
Random	70.341	70.178	70.196	63.168	55.767	
K-means	71.068	71.193	71.418	69.821	70.601	
Luminance less	72.365	77.216	69.917	63.948	56.709	
Luminance more	65.775	61.724	58.098	53.766	44.313	
Contrast less	71.749	70.679	69.437	62.825	55.455	
Contrast more	66.483	62.975	59.895	55.698	46.433	
FaceQNet	75.759	77.411	78.499	70.801	66.504	
Proposed FaceQNet mobile	75.984	77.948	79.267	72.028	68.082	
Proposed FaceQNet light	71.888	71.524	71.649	61.733	59.171	
Proposed FIIQA mobile	70.969	69.591	68.268	61.959	54.383	
Note:

The top three results are shown in bold.

Table 4 Recognition accuracy for IJB-C dataset, MobileNet-VGG2.

Algorithm	Number of selected frames K	
N/2	N/4	N/8	2	1	
Baseline		61.293	
Random	59.378	59.230	58.988	52.257	44.980	
K-means	62.102	62.108	61.503	61.759	61.185	
Luminance less	61.131	59.760	58.299	51.989	45.092	
Luminance more	56.314	51.673	47.937	43.284	35.004	
Contrast less	60.740	59.120	56.790	51.128	44.138	
Contrast more	56.963	52.726	49.411	44.787	36.482	
FaceQNet	65.127	65.631	66.532	59.689	54.532	
Proposed FaceQNet mobile	65.411	66.323	66.925	61.011	56.185	
Proposed FaceQNet light	61.373	60.537	59.313	50.751	47.525	
Proposed FIIQA mobile	59.259	57.516	55.900	50.349	43.552	
Note:

The top three results are shown in bold.

Table 5 Recognition accuracy for IJB-C dataset, Insightface.

Algorithm	Number of selected frames K	
N/2	N/4	N/8	2	1	
Baseline		69.834	
Random	69.871	70.182	69.835	63.899	62.048	
K-means	64.448	64.872	67.846	69.947	70.353	
Luminance less	67.449	63.701	62.787	57.771	55.095	
Luminance more	62.422	60.785	60.837	58.966	57.254	
Contrast less	67.893	63.906	62.682	57.895	55.962	
Contrast more	63.525	62.654	62.199	60.349	58.415	
FaceQNet	78.853	79.236	80.686	77.604	76.825	
Proposed FaceQNet mobile	79.428	79.852	80.706	78.957	77.623	
Proposed FaceQNet light	71.984	73.046	73.012	65.752	65.752	
Proposed FIIQA mobile	71.750	70.764	70.113	64.511	61.711	
Note:

The top three results are shown in bold.

The accuracy of face identification using the traditional approach for each frame on the IJB-C data set is 71% (Table 3), which is a high indicator due to the presence of many complex images. It is worth noting that performing keyframe searches using clustering methods works with approximately the same accuracy as Baseline. A number of algorithms for selecting high-quality video images have shown their effectiveness; the best-accuracy methods in the table are highlighted in bold. Based on the results, assessing the quality of frames using the FaceQNet model and its modifications increased the accuracy of facial identification by 7–9% compared to the traditional approach. Here, the maximum value of 79.267% is achieved through the FaceQNet mobile network, which was independently trained by the method of transferring knowledge. The LeNet-based FaceQNet light architecture yielded results slightly worse by 6–7%, however, the accuracy is still not inferior to the traditional approach. For these models, the optimal number of quality frames was revealed in the amount of 1/8 of the overall sequence N, where the average video duration is approximately 33 frames. Moreover, a further decrease in the sample leads to a sharp decrease in the accuracy of the result.

Recognition accuracy using less resource-intensive algorithms for calculating Brightness (1) and Contrast (3) is comparable to the traditional approach. However, the determination of the quality threshold in this case usually occurs empirically, which imposes significant limitations. The CNN model ResNet-50 shows better performance compared to MobileNet-VGG2 and InsightFace. Hover, this architecture is quite “cumbersome”. The accuracy of light MobileNet-VGG2 with FaceQNet mobile as frame quality assessment only 3% inferior to the Baseline of the deeper model ResNet-50.

The same experiments have been also conducted for the YTF dataset (Tables 6–8). The method of quality assessment using the light “FaceQNet mobile” architecture has shown its effectiveness. The accuracy of this algorithm exceeds the traditional approach by 3.3%. The selection of key frames through FIIQA mobile reaches Baseline in accuracy. Simple algorithms for estimating Brightness (1) and Contrast (2) are less effective for the studied data set.

Table 6 Recognition accuracy for YTF dataset, ResNet-50 (VGGFace2).

Algorithm	Number of selected frames K	
N/2	N/4	N/8	2	1	
Baseline		83.499	
Random	83.346	83.346	83.040	80.748	78.838	
K-means	83.499	83.575	83.575	83.728	83.498	
Luminance less	82.671	82.213	81.068	78.320	74.809	
Luminance more	76.121	75.828	74.803	72.708	70. 956	
Contrast less	82.977	82.061	80.611	78.473	75.725	
Contrast more	77.711	76.065	75.938	74.870	73.854	
FaceQNet	84.141	86.452	85.915	81.770	81.314	
Proposed FaceQNet mobile	85.397	86.249	86.790	82.877	80.658	
Proposed FaceQNet light	80.803	81.540	80.524	79.188	78.295	
Proposed FIIQA mobile	83.435	82.975	82.055	78.834	77.147	
Note:

The top three results are shown in bold.

Table 7 Recognition accuracy for YTF dataset, MobileNet-VGG2.

Algorithm	Number of selected frames K	
N/2	N/4	N/8	2	1	
Baseline		68.330	
Random	67.759	67.523	67.511	65.791	63.652	
K-means	69.440	68.994	68.228	68.101	68.002	
Luminance less	67.771	67.762	66.598	65.879	65.463	
Luminance more	62.059	61.942	61.779	60.943	59.487	
Contrast less	67.742	66.503	66.185	66.022	64.171	
Contrast more	63.313	63.217	60.832	60.779	59.050	
FaceQNet	70.988	71.882	71.750	69.880	68.706	
Proposed FaceQNet mobile	71.912	71.984	72.108	70.178	69.013	
Proposed FaceQNet light	69.984	68.649	69.127	67.071	66.921	
Proposed FIIQA mobile	68.668	67.786	67.315	66.810	66.882	
Note:

The top three results are shown in bold.

Table 8 Recognition accuracy for YTF dataset, Insightface.

Algorithm	Number of selected frames K	
N/2	N/4	N/8	2	1	
Baseline		80.854	
Random	79.198	79.082	79.001	77.996	75.823	
K-means	79.672	80.135	80.102	80.010	78.890	
Luminance less	78.467	77.265	77.245	76.624	74.058	
Luminance more	71.120	70.825	71.740	68.985	68.775	
Contrast less	79.532	78.984	77.416	77.261	75.379	
Contrast more	70.373	70.234	69.707	67.486	66.075	
FaceQNet	81.988	82.225	82.824	81.270	80.002	
Proposed FaceQNet mobile	81.792	83.323	84.002	80.654	80.325	
Proposed FaceQNet light	79.516	80.256	79.937	78.729	77.846	
Proposed FIIQA mobile	79.750	78.596	77.065	76.806	74.980	
Note:

The top three results are shown in bold.

It is worth noting that the best algorithms for the accuracy of the result are deep learning technologies, namely the FaceQNet model and its “light” modifications. In general, it was possible to achieve an increase in identification accuracy by 3-5% for all models for extracting feature vectors. Simple methods for estimating Brightness (1), Contrast (2), which work directly with the matrix of image pixels, turned out to be less effective.

Conclusion

This work is devoted to the study of effective methods for face identification by video based on the selection of high-quality frames. Both traditional methods for assessing the quality of faces based on Brightness (1), Contrast (2), and deep learning technology have been studied. We propose to train the lightweight CNN (“FaceQNet mobile”) for face quality analysis by distilling the knowledge of the FaceQNet ResNet-50 model. It was demonstrated that our key frame selection approach (Fig. 3) is much (up to 8–10%) more accurate when compared to conventional methods. It is important to emphasize that the usage of our distilled model leads to even higher accuracy when compared to original FaceQNet model. It has been experimentally shown that the use of fast methods for evaluating the quality of image leads to a decrease in the time of direct passage through a CNN due to a reduction in the set of frames under consideration.

In future studies, it will be necessary to study the methods for adaptive selection of the number of key frames K. Indeed, complex videos usually needs more frames to reliably identify an observed subject, though even one key frame may be enough for the simplest input videos without noice, occlusion, etc. One possibility here is to apply sequential analysis that was used previously to improve the speed of image recognition (Savchenko, 2019b). Moreover, it is necessary to apply the same techniques for other video-based face processing tasks including age/gender/ethnicity/emotion prediction (Kharchevnikova & Savchenko, 2018; Savchenko, 2019a).

Additional Information and Declarations

Competing Interests

Author Contributions

Data Availability

The authors declare that they have no competing interests.

Angelina Kharchevnikova conceived and designed the experiments, performed the experiments, analyzed the data, performed the computation work, prepared figures and/or tables, authored or reviewed drafts of the paper, and approved the final draft.

Andrey V. Savchenko conceived and designed the experiments, analyzed the data, prepared figures and/or tables, authored or reviewed drafts of the paper, and approved the final draft.

The following information was supplied regarding data availability:

The KeyFrameExtraction is available at GitHub:

https://github.com/KhAngelina/KeyFrameExtraction.

Raw data and trained models described in the article are also available at GitHub: https://github.com/KhAngelina/KeyFrameExtraction/tree/master/models.

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
