# Peer review of "Efficient video face recognition based on frame selection and quality assessment"

_PeerJ Computer Science, doi:10.7717/peerj-cs.391_

## Round 0.1 · original submission · Major Revisions

This paper needs further revision.

Reviewer 1 ·

Basic reporting

1. The writing of the paper (e.g. grammar, tense) should be improved to make it more professional.
2. Some references on related works including gait recognition and quality assessment are suggested to be added for sufficient field background.
[1] Xianye Ben, Peng Zhang, Zhihui Lai, Rui Yan, Xinliang Zhai, Weixiao Meng. A general tensor representation framework for cross-view gait recognition, Pattern Recognition, vol. 90, pp. 87-98, 2019.
[2] Lei Chen and Jiying Zhao, No-reference perceptual quality assessment of stereoscopic images based on binocular visual characteristics, Signal Processing: Image Communication, vol. 76, pp. 1-10, 2019.
3. The format of references should be indentical, e.g. Line 332 and Line 340.

Experimental design

Many important details are missing in the experimental design. For instance, how many dataset (or proportion) is used for training or testing? Some parameters of Adam optimizer are also not provided.

Validity of the findings

1. For the compared methods, Are the results obtained from the corresponding papers or conducted in your experiments, which need to be clarified.
2. More explanations or analysis should be added for performance comparion, instead of just listing the results.

Additional comments

The paper proposed an efficient video face recognition based on frame selection and
quality assessment. There are several issues to be addressed based on the comments.

Reviewer 2 ·

Basic reporting

Your method needs more detail. I suggest that you improve the description of the proposed method to your research more clear to readers

Experimental design

Experiment is Ok.

Validity of the findings

This article has certain innovation.

Additional comments

The authors train the lightweight CNN (“FaceQNet mobile”) for face quality analysis by distilling the knowledge of the FaceQNet ResNet-50 model. The reported experiments demonstrate the performance of the proposed method. My comments are as follows:
1. The innovation of this paper is not clear and it is difficult for readers to understand the main contributions of this paper. This part should be added in Introduction section.
2. In the proposed framework (Figure 3), which part is the method proposed by the authors?
3. In Figure 1, what is teacher model? and what is Student model?
4. Some details of the proposed method are not clearly described in the paper.

---

## Round 0.2 · Minor Revisions

The authors have tried to answer most of the questions. The name and surname of some references are not correct. Please check them.

Reviewer 2 ·

Basic reporting

no comment

Experimental design

no comment

Validity of the findings

no comment

Additional comments

This reviewer would like to thank the authors for their effort in clarifying and modifying this paper. I have no other questions. I suggest that the paper is accepted for publication.

---

## Round 0.3 · accepted · Accept

This paper can be accepted.